# Correlations between Maternal and Fetal Outcomes in Pregnant Women with Kidney Failure

**DOI:** 10.3390/jcm12030832

**Published:** 2023-01-20

**Authors:** Daniela-Catalina Meca, Valentin Nicolae Varlas, Claudia Mehedințu, Monica Mihaela Cîrstoiu

**Affiliations:** 1Doctoral School of “Carol Davila” University of Medicine and Pharmacy, 4192910 Bucharest, Romania; 2Faculty of Medicine, “Carol Davila” University of Medicine and Pharmacy, 050451 Bucharest, Romania; 3Department of Obstetrics and Gynecology, Filantropia Clinical Hospital, 011171 Bucharest, Romania; 4Department of Obstetrics and Gynecology, University Emergency Hospital Bucharest, 050098 Bucharest, Romania

**Keywords:** acute renal failure, chronic kidney disease, pregnancy outcome

## Abstract

Background: Kidney function impairment in pregnancy is challenging, with incidence and prognosis only partially known. Studies concerning maternal and fetal outcomes in pregnancies occurring in patients with renal injury and the therapeutic strategies for improving the prognosis of these patients are scarce due to the limited number of cases reported. Objectives: We aimed to establish correlations between the main maternal and fetal outcomes in patients with severe CKD or AKI in pregnancy to improve the prognosis, referring to a control group of patients with mild kidney impairment. Methods: For this purpose, we conducted a retrospective study, at University Emergency Hospital in Bucharest, Romania, from January 2019 until December 2021, selecting 38 patients with AKI and 12 patients diagnosed with advanced CKD, compared to 42 patients displaying borderline values of serum creatinine (0.8–1 mg/dL), reflecting the presence of milder kidney impairment. Results: The probability of having a child that is premature and small for gestational age, with a lower Apgar score and more frequent neonatal intensive care unit admissions, delivered by cesarean section, is higher in patients with highly reduced kidney function. Conclusion: Severe kidney function impairment is associated with a grim fetal prognosis and obstetrical complications.

## 1. Introduction

During pregnancy, anatomical and physiological changes occur in all maternal organs and systems to support the metabolic requirements necessary for the fetus’s survival and development. Thus, the kidneys increase in size by 1–1.5 cm, and the renal volume increases by 30%. As a result of systemic vasodilatation and volume expansion, the glomerular filtration rate increases by 40–50%, reaching 180 mL/minute in the perinatal period. Consequently, creatinine clearance increases by 40–50% with reduced serum creatinine. In addition, there is increased urinary excretion of uric acid, glucose, proteins, and albumin. In contrast, there is reduced reabsorption of amino acids and beta-microglobulin. All these adaptive changes may make it difficult to diagnose renal injury promptly [1].

Renal injury associated with pregnancy, including structural and functional renal alterations, is challenging [2,3,4]. Its incidence is difficult to assess due to the interference of physiological changes [2,5]. Furthermore, there is no consensus on the definition of acute and chronic renal failure in pregnancy [6].

With a reported incidence of 4.5 per 10,000 deliveries, acute kidney injury (AKI) is a heterogeneous situation that can develop at any perinatal time [7,8,9]. Considering the trigger factors of AKI, a bimodal distribution is usually described: an early peak representing an outcome of severe hemorrhage, septic abortion, or hyperemesis gravidarum and a late peak representing obstetric complications, such as hypertensive disorders of pregnancy, placenta previa, acute fatty liver, and postpartum hemorrhage [3]. The incidence of pregnancy-related AKI is modulated by other factors, such as advanced maternal age, assisted reproduction techniques, and obesity. Obstetric causes of AKI are reported in <1 in 20,000 pregnancies [7,8,9].

The American College of Obstetricians and Gynecologists defines pregnancy-related AKI as an increase in serum creatinine level over 1.1 mg/dL [10,11].

Currently, a classification (RIFLE, AKIN) that defines the degrees of severity of AKI in pregnancy, as well as the maternal outcomes dependent on the loss of renal function, has not been well confirmed, most likely due to the added physiological changes [12].

The following criteria are used to diagnose virtually all reported AKI cases: a creatinine level ≥1 mg/dL, a sudden increase of 0.5 mg/dL from baseline in less than 48 h, oliguria or anuria, or the need to institute dialysis [6,13].

Preeclampsia is the hallmark of hypertensive disorder in pregnancy, with an increased risk of developing chronic kidney disease (CKD) [14].

CKD is reported in up to 3–5% of pregnancies [15,16,17]. Although increased maternal age increases the prevalence of CKD, the standard definition, described as a decrease in glomerular filtration rate under 60 mL/min for a period of minimum 3 months, is not applicable in pregnancy due to interference with pregnancy-related hyperfiltration [18,19]. Kidney disease, including glomerulonephritis, lupus erythematous systemic, chronic pyelonephritis, and diabetic nephropathy, may further modulate the risk of adverse pregnancy outcomes [20].

AKI and CKD increase the risk of maternal and fetal complications, including preeclampsia, stillbirth, small for gestational age (SGA), fetal growth restriction (FGR), preterm birth, and cesarean section, and they may lead to a further decline in renal function [11,21]. The outcomes for women with CKD depend on the degree of renal dysfunction [22].

This study aims to evaluate the effect of maternal CKD and AKI on maternal–fetal outcomes compared to a control group of patients with mild renal impairment.

## 2. Materials and Methods

We performed a retrospective study to evaluate the maternal and fetal outcomes in women with acute and chronic renal disease, and we compared their outcomes with those observed in women with milder kidney impairment (serum creatinine values 0.8–1 mg/dL) over 3 years. All parameters were recorded at hospital admission, and reference values were those of the central laboratory of the University Emergency Hospital in Bucharest, Romania. All CKD patients included were diagnosed with CKD stages 3–5 before hospital admission or pregnancy. None of the study pregnancies were obtained through assisted reproduction techniques.

### 2.1. Study Design

We performed a retrospective study on women with acute and chronic renal diseases in pregnancy observed over 3 years (from January 2019 to December 2021). The study was conducted at the University Emergency Hospital in Bucharest, Romania, an obstetric tertiary care unit treating high-risk pregnancies. Throughout the period of study, there were 6707 deliveries. We collected and analyzed data from 50 pregnant women with AKI or advanced CKD and 42 cases of pregnant women with milder kidney function impairment.

We included all the cases with AKI, regardless of the trigger factor: sepsis, hypovolemia, and hypertension with its complications. For CKD, we included all the pregnant women with this diagnosis in the study, regardless of the underlying disease: lupus erythematous systemic, glomerulonephritis, chronic pyelonephritis, and polycystic renal disease. Only the patients with glomerulonephritis had a renal biopsy performed; the other diagnoses were established in the nephrology departments based on laboratory findings and imaging.

The exclusion criteria were as follows: the absence of informed consent, pregnant women under 16 years, AKI caused by septic abortions in the first trimester, or hemorrhage from extrauterine pregnancies.

The cutoff of 1.19 mg/dL of serum creatinine used in this study was established following ROC curve analysis.

The patient distribution is shown in Figure 1.

### 2.2. Data Collection

The baseline data were obtained from hospital observation sheets and the patient’s medical files. The patient’s informed consent was a mandatory inclusion criterion. The identification data were not included in the database we developed, which is structured as follows:

-Personal data: age and area of residence.

-Obstetric/gynecological history: abortions/pregnancies and ante/perinatal complications.

-Details related to the current pregnancy: pregnancy follow-up or not, gestational age (GA) at the time of assessment, pregnancy-related disorders (hypertension, gestational diabetes, anemia, thrombocytopenia, and proteinuria), GA at which renal failure was diagnosed, and the mode of delivery (cesarean section/vaginal birth).

-Data regarding fetal parameters: complications during pregnancy (SGA/FGR), fetal birth weight, 1 min Apgar score, and neonatal intensive care admission.

-Data regarding the underlying disease that caused the CKD and the current treatment (hemodialysis) as the trigger factors for AKI.

-Data on the evolution of biochemical parameters (values of nitrogen retention on admission and its evolution during hospitalization).

-The need for dialysis during hospitalization.

### 2.3. Statistics

Data analysis was performed using SPSS 26.0 Software. The Kolmogorov–Smirnov analysis was performed to assess the normal data distribution through the abovementioned variables. The Pearson correlation test was performed to establish the impact of risk parameters (high serum creatinine values) on different variables (fetal weight, Apgar score, and GA at delivery). The chi-square test was applied to establish the interdependence between nominal variables. Results with *p* < 0.05 were statistically significant.

## 3. Results

During the 3 years of the study, out of 6707 registered births, 92 cases were eligible for our study: 50 cases of acute and chronic severe kidney impairment and 42 cases with borderline values of serum creatinine, indicative of milder kidney function impairment. Thus, over this period, the incidence was 0.56% of pregnancy-related AKI, and advanced CKD was present in 0.17% of the cases, while 0.62% had a mild increase in serum creatinine. The average maternal age was 30.87 ± 6.9 (18–42 years) for the AKI group and 29 ± 6.49 (21–42 years) for the CKD group. In the patients with milder serum creatinine values, the mean age was 31.23 ± 5.88 (18–42 years) (Figure 2).

### Description of the Groups

The patients included in the study were divided into three groups:(1)AKI group *n* = 38;(2)CKD group *n* = 12 with late-stage disease (stages 3–5);(3)Control group with a milder increase in serum creatinine level *n* = 42 (Table 1).

The gestational age (GA) at delivery revealed a risk of preterm birth in all three groups. In the AKI group, the average GA at delivery was 33.79 ± 4.73 weeks, with an increase of almost 2 weeks in the borderline serum creatinine values group (35.52 ± 4.03 weeks). In contrast, there was not a significant difference between the GA at delivery in the groups with severe acute and chronic kidney impairment (33.79 vs. 34.42 weeks).

In the control group, we included seven pregnancies with no record of prenatal care who had a mean value of serum creatinine at the admission of 0.88 ± 0.07 mg/dL (95% CI 0.82 to 0.93), similar to that observed in the patients who benefited from prenatal care (0.86 ± 0.05 mg/dL) (95% CI 0.84 to 0.87).

In the AKI group, we enrolled nine pregnancies with no record of prenatal care, with a mean serum creatinine value at the admission of 1.97 ± 1.13 mg/dL (95% CI 1.29 to 2.66). The mean value of serum creatinine in the AKI pregnancies that benefited from prenatal care was 1.55 ± 1,24 mg/dL (95% CI 1.06 to 2.04), with no significant difference compared with the previous group.

In the advanced CKD group, three out of the 12 patients included were pregnancies without prenatal care, despite their known history of kidney injury, with serum creatinine values at the admission of 0.65 mg/dL, 1.95 mg/dL, and 8.98 mg/dL.

During pregnancy, the considered normal range of creatinine concentration varies between 0.40–0.80 mg/dL, with values higher than 0.8 mg/dL raising the suspicion of possible kidney damage [23]. As a result, in our study, the range of 0.8–1 mg/dL of the control group corresponded to mild renal lesions, which can frequently be underdiagnosed. The control group with a “borderline value of serum creatinine” of 0.8–1 mg/dL presumably included patients with CKD stages 1–3 before pregnancy.

The prognosis was unfavorable for the patient with 8.98 mg/dL serum creatinine at admission, who needed to start continuous renal replacement therapy in the ICU. The patient with 1.95 mg/dL serum creatinine died due to macrophage activation syndrome associated with lupus nephritis.

Anemia was diagnosed in 63.15% of AKI cases and 58.33% of patients in the CKD group. Furthermore, 50% of CKD patients and 36.84% of AKI patients underwent dialysis, with most AKI patients requiring only short-term dialysis in the ICU.

The underlying causes of CKD and the trigger factor of AKI are reported in Table 2.

The maternal prognosis of the dialysis patients was favorable, with complete restoration of renal function in only four cases (28.57%) in the AKI group and one case (16.66%) in the CKD group (Table 3 and Appendix A).

Considering the higher percentage (16%) of patients who died in the study groups, in Table 4, we summarized the evolution of these patients intending to adopt future therapeutic schemes to reduce the mortality rate.

The relationship between hospitalization days and serum creatinine at admission was significant only in the AKI group (*p* = 0.01). The value of serum creatinine at admission was not a predictive factor for the fetal outcome or admission to NICU.

Considering a threshold value for serum creatinine of 1.12 mg/dL, we found that, in the AKI group, the fetal weight was similar, and the Apgar score was higher in the group with serum creatinine levels <1.12 mg/dL. The number of stillbirths was equal in both subgroups. In the CKD group, the fetal weight was significantly higher in the subgroup with serum creatinine levels <1.12 mg/dL (2165 g vs. 1875 g, *p* = 0.03). A favorable maternal result was recorded in almost all cases with creatinine <1.12 mg/dL (93.33% in the AKI group and 100% in the CKD group), while, in the CKD patients with serum creatinine levels >1.12 mg/dL, 37.5% needed dialysis (Table 5).

Considering the limit of 35 mg/dL of serum urea, we identified a difference among the three subgroups regarding fetal outcomes. In the borderline group, in patients with serum urea >35 mg/dL, an increased rate of low-birth-weight newborns was observed compared to those from patients with values <35 mg/dL (*p* = 0.006).

The risk of stillbirth was significantly higher in the subgroup of AKI with serum urea >35 mg/dL versus the cases with serum urea <35 mg/dL (Table 6).

## 4. Discussion

Renal diseases encountered during pregnancy can get worse, and the start of dialysis may be necessary to reduce maternal and perinatal morbidity and mortality rates. Major discrepancies regarding patients’ access to dialysis services are evident in low- and middle-income countries [3]. As mentioned above, the incidence of pregnancy-related AKI in our study was 0.56%, in keeping with the meta-analysis by Trakarnvanich et al., which reported an overall incidence of 2%, ranging from 0.14% in developed countries to 4.11% in developing countries [24]. The incidence was reported to have a 10% yearly increase, with a rise from 2.4 to 6.3 per 10,000 deliveries between 1999 and 2011 in a study by Mehrabadi et al. in the United States [7]. The etiology most frequently involved was preeclampsia in 42.1%, followed by abruptio placentae in 15.8% of cases, which is similar to a study performed by Pahwa et al. on 27 patients with postpartum AKI (40.7% of cases were secondary to preeclampsia) [25]. The chances of adverse events, including preeclampsia and preterm birth, depending on the different kidney function impairment stages.

Regardless of the underlying cause of pregnancy-related AKI, Liu et al., in 2017, reported a correlation between preeclampsia with lower mean GA at delivery and lower fetal birth weight [26]. In our study, in patients with preeclampsia, the mean GA at delivery was 34.87 weeks with a mean birth weight of 2096.84 g. Although the prevalence of preeclampsia was increased in the control group, which could cause renal dysfunction, all cases had a favorable evolution from the point of view of renal function, without the need for dialysis.

Pregnancy may be the first occasion for diagnosing pre-existing kidney disease, occurring in 23.77% of our cases. The value of serum creatinine at admission was not correlated with poor fetal outcomes. Preterm birth in the AKI group (65.78%) was caused by obstetrical complications.

A study performed by Luders et al. emphasized that blood urea nitrogen <35 mg/dL was associated with an improved fetal prognosis in patients on hemodialysis [27]. When analyzing that parameter, we found that the number of stillbirths was higher in patients with serum urea >35 mg/dL (four stillbirths) compared to that with serum urea <35 mg/dL (two stillbirths). Compared to the study performed on 471 pregnancies with AKI, which recorded 27% of stillbirths, our study recorded only 15.8% [28].

In the AKI group, the maternal outcome in patients with serum creatinine >1.12 mg/dL was favorable in 73.91% of cases; meanwhile, in patients with serum creatinine values <1.12 mg/dL was favorable in 93.33% of all patients.

A study performed by Mir et al. on 28 patients with postpartum AKI who required dialysis showed a higher percentage (53.5%) of favorable maternal outcomes, with complete restoration of renal function [29]. The maternal mortality in the AKI group was 18.42%, similar to that described by Goplani et al. on a group of 70 pregnant women with AKI caused by obstetrical complications (18.57%) [30].

Regarding maternal outcome, we found a relationship between the serum creatinine value at admission and the duration of hospitalization (*p* = 0.01). The maternal outcome was favorable in 94.73% (18/19) of all cases with serum creatinine <1.12 mg/dL, while, above this threshold, 10 patients (43.47%) from the AKI group and five patients (62.5%) from the CKD group needed dialysis, and the hospitalization period was prolonged, with a mean of 13.5 days. The longer hospitalization period was associated with worsening renal function following a study published by Shah et al. [31].

In our study, the prevalence of patients with advanced CKD was 0.17%, while other studies noted a prevalence of 1.2% [15,16,17]. The etiology is similar to that in the general population and other studies on advanced CKD: lupus erythematous systemic, glomerulonephritis, and chronic pyelonephritis [20].

The 1.12 mg/dL cutoff of serum creatinine used in this study and adapted to all types of renal lesions is close to the threshold value of 1.24 mg/dL described in a study by Jungers et al. on a group of 360 patients with mild chronic renal dysfunction, and supported by Williams et al. In patients whose serum creatinine level remained below the aforementioned threshold, no adverse effects on renal function were described, with a favorable maternal outcome [32,33], as in our study.

Adverse maternal and fetal outcomes are confirmed as correlated with the CKD stage. Elevated creatinine levels in the advanced CKD group predispose to a high risk of preterm birth and low fetal birth weight [34].

The therapeutic management underlined how the need for dialysis was associated with unfavorable maternal prognosis. In our study, half of the pregnant women with CKD who required dialysis had a significant risk of preterm birth and NICU admission results similar to those described in a study by Zoler, who confirmed that pregnant women on dialysis had a significant risk for preterm births (85%) and NICU admission (67%) [35].

All patients with CKD in an advanced stage needed a cesarean section, in keeping with the study by Lim et al., which reported a cesarean section rate of 42.9% in early-stage patients (stages 1–2) and 100% in late-stage patients (stages 3–5) [36].

Changes related to glomerular rate filtration can be underestimated during pregnancy; therefore, it is suggested that a serum creatinine level above 0.87 mg/dL should be considered abnormal and require further investigation [18,19,37]. Thus, we analyzed a group of patients with serum creatinine within 0.8–1 mg/dL. This group was characterized by a risk of preterm birth of 35.71%, which is lower than that in the AKI and advanced CKD groups. Furthermore, the risk of SGA was 42.85%, and the risk for NICU admission was 42.5%, both values being lower than in the AKI group.

Our study had several limitations, mainly the reduced number of cases enrolled. Furthermore, we did not have access to kidney function parameters during each trimester of pregnancy and after discharge.

## 5. Conclusions

CKD and AKI in pregnancy are confirmed as associated with a poor maternal and fetal prognosis. AKI during pregnancy continues to represent a major public health problem. Early identification of risk factors (advanced maternal age, assisted reproductive techniques, comorbidities, and hemorrhages) may improve early diagnosis and therapeutic management. Monitoring renal function in pregnant women with CKD is essential because there is a major risk of worsening renal function and of serious pregnancy complications (preterm birth, stillbirths, small-for-gestational-age newborns, and preeclampsia).

## Figures and Tables

**Figure 1 jcm-12-00832-f001:**
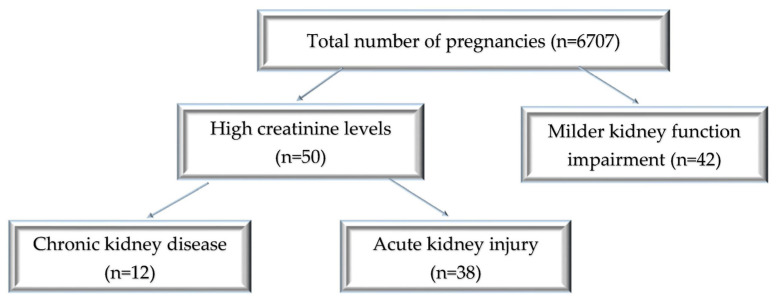
Diagram of patient distribution according to the type of renal injury and the values of creatinine levels.

**Figure 2 jcm-12-00832-f002:**
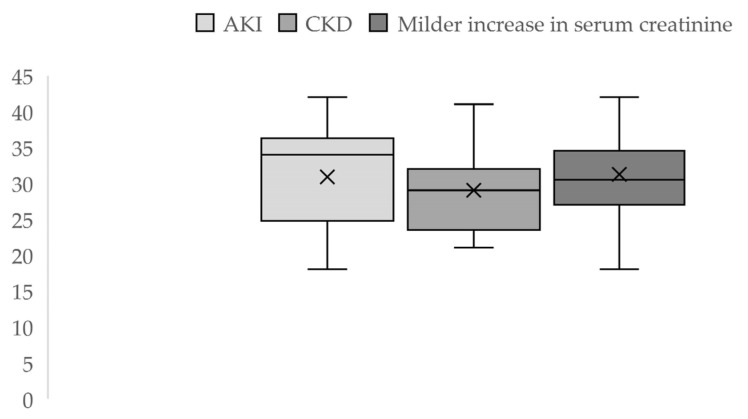
Descriptive statistics of maternal age by groups.

**Table 1 jcm-12-00832-t001:** Clinical features and maternal/neonatal outcomes.

	AKI	Advanced CKD	Milder Increase of Serum Creatinine	*p*-Value *
Demographic data	*n* = 38	*n* = 12	*n* = 42	
Maternal age, years (mean *±* SD)	30.87 *±* 6.90	29.00 *±* 6.49	31.23 *±* 5.88	0.58
Area of residence				
Urban (n, %)	20 (52.63%)	8 (66.66%)	29 (69.04%)	0.30
Rural (n, %)	18 (47.36%)	4 (33.33%)	13 (30.96%)	
Parity				
Primiparous (n, %)	23 (60.52%)	8 (66.66%)	29 (69.04%)	0.72
Multiparous (n, %)	15 (39.48%)	4 (33.33%)	13 (30.96%)	
Investigated pregnancy (n, %)				
Yes	29 (76.32%)	9 (75%)	35 (83.34%)	0.15
No	9 (23.68%)	3 (25%)	7 (16.66%)	
Maternal outcomes & clinical features				
Preeclampsia	16 (42.1%)	8 (66.67%)	33 (78.6%)	0.339
Anemia (*n*, %)	24 (63.15%)	7 (58.33%)	12 (28.57%)	0.58
Dialysis (*n*, %)	14 (36.84%)	6 (50%)	-	0.571
Maternal mortality	7 (18.42%)	1 (8.33%)	-	0.007
Mode of delivery				
Cesarean section	36 (94.74%)	12 (100%)	42 (100%)	0.16
Vaginal birth	2 (5.26%)	-	-	
Days of hospitalization (mean *±* SD)	12.78 *±* 9.07	20.75 *±* 26.16	6.38 *±* 7.26	0.001
Hemoglobin, g/dL (mean *±* SD)	9.93 *±* 2.26	10.26 *±* 2.45	11.6 *±* 1.79	0.806
Serum creatinine at admission, mg/dL (mean *±* SD)	1.70 *±* 1.21	3.23 *±* 2.88	0.87 *±* 0.05	0.045
Serum creatinine at discharge, mg/dL (mean *±* SD)	1.26 *±* 0.85	2.95 *±* 2.35	0.71 *±* 0.13	0.915
Serum urea at admission, mg/dL (mean *±* SD)	50.68 *±* 28.66	65.78 *±* 46.20	26.50 *±* 9.29	0.084
Serum urea at discharge, mg/dL (mean *±* SD)	52.84 *±* 30.84	84.81 *±* 58.12	28.79 *±* 8.90	0.752
Proteinuria, mg/L (mean ± SD)	261.31 *±* 83.18	244.16 *±* 102.46	226.66 *±* 106.58	0.092
eGFR, mL/min/1.73 m^2^ (mean ± SD)	52.26 ± 34.91	43.49 ± 42.26	76.60 ± 6.79	0.690
Neonatal outcomes				
Gestational age (weeks) (mean *±* SD)	33.79 *±* 4.73	34.42 *±* 2.9	35.52 *±* 4.03	0.18
Birth weight (grams) (mean *±* SD)	2096.84 *±* 836.39	1971.66 *±* 565.45	2628.57 *±* 946.82	0.009
Preterm birth (*n*, %)	25 (65.78%)	8 (66.67%)	15 (35.71%)	0.809
SGA (*n*, %)	16 (42.10%)	11 (91.11%)	18 (42.85%)	0.522
Stillbirth (*n*, %)	6 (15.78%)	-	2 (4.76%)	0.22
1 min Apgar score (mean *±* SD)	6.50 *±* 2.56	7.92 *±* 1.16	7.95 *±* 1.33	0.005
NICU admission (*n*, %)	21 (55.26%)	10 (83.33%)	17 (42.5%)	0.06

* *p* < 0.05 was considered statistically significant; eGFR—estimated glomerular filtration rate, SGA—small for gestational age, NICU—neonatal intensive care unit.

**Table 2 jcm-12-00832-t002:** Distribution of underlying pathology/trigger factors.

Acute Kidney Injury	Cases	Percentage
Preeclampsia	16	42.1
Abruptio placentae	6	15.8
Hemorrhage	4	10.5
Eclampsia	3	7.9
Urosepsis	2	5.3
HELLP syndrome	2	5.3
SARS-CoV-2 infection	1	2.6
Scleroderma crisis	1	2.6
Metastatic breast neoplasm	1	2.6
Generalized anasarca	1	2.6
Diabetes mellitus	1	2.6
Total	38	100.0
Chronic kidney disease		
Membranous glomerulonephritis	3	25.0
Lupus erythematous systemic	3	25.0
Polycystic kidney	3	25.0
Chronic pyelonephritis	2	16.7
Tubular acidosis	1	8.3
Total	12	100.0
Milder kidney function impairment		
Preeclampsia	33	78.6
Abruption placentae	3	7.1
SARS-CoV-2 infection	3	7.1
HELLP syndrome	1	2.4
Hemorrhage	1	2.4
Eclampsia	1	2.4
Total	42	100.0

**Table 3 jcm-12-00832-t003:** Maternal outcomes in dialysis patients related to the serum creatinine threshold value.

	Serum Creatinine <1.12 mg/dL		Serum Creatinine >1.12 mg/dL	
	AKI Group (*n* = 4)	CKD Group (*n* = 1)	AKI Group (*n* = 10)	CKD Group (*n* = 5)
Favorable (*n*, %)	3 (75%)	1 (100%)	4 (40%)	4 (80%)
Necessary hysterectomy	-	-	2 (20%)	1 (20%)
Pulmonary edema	-	-	1 (10%)	-
MODS	2 (50%)	-	1 (10%)	-
Dialysis				
- Continuous *	-	-	-	3 (60%)
- Short-term	4 (100%)	1 (100%)	10 (100%)	2 (40%)
Exitus	1 (25%)	-	6 (60%)	1 (20%)

* continuous renal replacement therapy.

**Table 4 jcm-12-00832-t004:** Synopsis of maternal deaths among groups.

Cases	Age	Group	Serum Creatinine at Admission	Last Serum Creatinine Value	Renal Impairment	Comorbidities	Therapeutic Strategy in ICU	Short- Term Dialysis	Neonatal Outcome	Maternal Outcome
1	35	AKI	1.23	0.98	Unknown prior admission	SARS-CoV-2 Pneumonia	OTI, + inotropes, AH, albumin, antibiotics, AC, BD	+	stillbirth	MODS
2	35	AKI	1.12	2.03	Unknown prior admission	HELLP syndrome	OTI, + inotropes, AH, albumin, antibiotics, AC, BD	+	stillbirth	MODS
3	38	AKI	1.92	0.54	Unknown prior admission	Abruptio placentae	OTI, + inotropes, AH, albumin, antibiotics, AC, BD, blood transfusion	+	stillbirth	MODS
4	18	AKI	1.29	2.82	Unknown prior admission	Eclampsia	OTI, + inotropes, AH, albumin, antibiotics, AC, BD	+	stillbirth	Maternal death
5	36	AKI	2.55	1.58	Unknown prior admission	Scleroderma crisis	OTI, + inotropes, AH, albumin, antibiotics, AC, BD	+	Apgar score = 7 BW = 700 g	Maternal death
6	34	AKI	4.43	1.05	Unknown prior admission	Hemorrhagic shock (Placenta percreta)	OTI, + inotropes, albumin, antibiotics, AC, BD, blood transfusion	+	Apgar score = 8 BW = 3200 g	Cardiac arrest through asystole
7	37	AKI	1.27	3.02	Unknown prior admission	Metastatic breast cancer	OTI, + inotropes, albumin, antibiotics, AC, BD	+	Apgar score = 3 BW = 640 g	Cardiac arrest through asystole
8	21	CKD	1.95	1.96	Lupus nephritis	-	OTI, + inotropes, AH, albumin, antibiotics, AC, BD	+	Apgar score = 6 BW = 1100 g	Macrophagic activation syndrome

OTI—orotracheal intubation; AH—antihypertensive drugs; AC—anticoagulants; BD—bronchodilators; +—positive; MODS—multiple organ dysfunction systems; BW—birth weight.

**Table 5 jcm-12-00832-t005:** Neonatal and maternal characteristics by a threshold value of serum creatinine.

	Serum Creatinine < 1.12 mg/dL			Serum Creatinine >1.12 mg/dL		
	AKI Group (*n* = 15)	CKD Group (*n* = 4)	*p*-Value	AKI group (*n* = 23)	CKD Group (*n* = 8)	*p*-Value
Neonatal outcome						
Birth weight (g)	2162.66 ± 1025.89	2165 ± 413.48	0.402	2053.91 ± 708.45	1875 ± 630.19	0.262
Apgar score	7.41 ± 0.16	8 ± 0.81	0.079	5.66 ± 3.27	7.87 ± 1.35	0.705
Stillbirth	3 (20%)	-	0.53	3 (13.04%)	-	0.39
Maternal outcome						
Favorable (*n*, %)	14 (93.33%)	4 (100%)		17 (73.91%)	7 (87.5%)	
Necessary hysterectomy	1 (6.66%)	-		4 (17.39%)	1 (12.5%)	
Pulmonary edema	-	-		1 (4.34%)	-	
MODS	2 (13.33%)	-		2 (8.69%)	-	
Dialysis						
- Continuous*	-	-		-	3 (37.5%)	
- Short-term	4 (26.66%)	1 (25%)		10 (43.47%)	2 (25%)	
Exitus	1 (6.66%)	-		6 (26.08%)	1 (12.5%)	

* Continuous renal replacement therapy.

**Table 6 jcm-12-00832-t006:** Neonatal characteristics by a threshold value of serum urea.

	Serum Urea under 35 mg/dL				
	Birth Weight (g) (Mean ± SD)	*p*-Value	Apgar Score (Mean ± SD)	*p*-Value	Stillbirth (*n*)
AKI group	2138.89 ± 811.51	0.07	7 ± 2.38	0.28	2
CKD group	2087.5 ± 643.46		7.75 ± 0.7		0
Control group	2789 ± 893.39		8.02 ± 1.32		1
	Serum urea over 35 mg/dL				
	Birth weight (g) (mean ± SD)		Apgar score (mean ± SD)		Stillbirth (*n*)
AKI group	2083.79 ± 842.27	0.54	6.36 ± 2.64	0.19	4
CKD group	1913.75 ± 574.69		8 ± 1.22		0
Control group	1665 ± 866.98		7.02 ± 1.3		1

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
