# Peer review of "Correlations between Maternal and Fetal Outcomes in Pregnant Women with Kidney Failure"

_jcm, 2023, doi:10.3390/jcm12030832_

Round 1
Reviewer 1 Report
Dear Authors you need to make some changes
Introduction
Line 37, the word “peripartum” must be changed. Do you mean “perinatal”?
You make little mention of pre-eclampsia which is very common in pregnancy.
Also, the mechanism of renal function and the differentiation in pregnancy is not mentioned.
In general the Introduction is not well supported.
Methods
What is defined as acute and what as chronic renal failure?
All conditions had the same weight to be included in the study?
Was co-morbidity, maternal weight or assisted reproductive methods considered?
Limitations
We do not know in all cases the values of laboratory tests before pregnancy.
Conclusions
There are many categories of kidney failure, your conclusion should be more specific without such simplistic generalizations
Author Response
Reply to Reviewer 1
Introduction
Line 37, the word “peripartum” must be changed. Do you mean “perinatal”?
Answer: Thank you for your suggestion; we changed it. (please see the attached manuscript) Line 49.
You make little mention of preeclampsia which is very common in pregnancy.
Answer: Thank you for your valuable remarks; we added more information. (please see the attached manuscript) Lines 58-64.
Also, the mechanism of renal function and the differentiation in pregnancy is not mentioned.
Answer: Thank you for your comments; we added the mechanism of renal function.
(please see the attached manuscript) Lines 31-41.
In general, the Introduction is not well supported.
Answer: Thank you for your valuable recommendation; we structured the introduction for a better understanding. (please see the attached manuscript)
Methods
What is defined as acute and what as chronic renal failure?
All conditions had the same weight to be included in the study?
Was co-morbidity, maternal weight or assisted reproductive methods considered?
Answer: Thank you for your valuable remarks; we added more data according to your recommendations. (please see the attached manuscript) Lines 106-119.
Limitations
We do not know in all cases the values of laboratory tests before pregnancy.
Answer: Thank you for your mention; we added this in the limitations section. (please see the attached manuscript) Lines 451-452.
Conclusions
There are many categories of kidney failure, your conclusion should be more specific without such simplistic generalizations.
Answer: Thank you for your recommendation; we rewrite the entire section to be more precise. (please see the attached manuscript). Lines 454-466.
Kindest regards
Reviewer 2 Report
The topic of kidney failure in pregnancy/induced by pregnancy is always interesting. The authors presented results from a tertiary hospital in Romania, which is epidemiologically valuable. The main problem is that the text is confusingly written, the groups (it cannot be only serum creatinine level) and the outcomes are not clearly qualified. The correlations of some indicators, for example the number of days of hospitalization, do not seem important. The manuscript should be shortened and focused on the main results, that these patients delivered mostly by caesarean section, newborns are premature and small for gestational age, have lower Abgar score, and that dialysis is associated with a poor outcome.
Author Response
Reply to Reviewer 2
The topic of kidney failure in pregnancy/induced by pregnancy is always interesting. The authors presented results from a tertiary hospital in Romania, which is epidemiologically valuable.
The main problem is that the text is confusingly written, the groups (it cannot be only serum creatinine level), and the outcomes are not clearly qualified.
Answer: Thank you for your valuable remarks; we added more information and restructured the results. (please see the attached manuscript) Lines 174-309.
The correlations of some indicators, for example the number of days of hospitalization, do not seem important.
Answer: Thank you for your comments; we discuss this correlation according to the AKI group.
(please see the attached manuscript) Lines 267-272.
The manuscript should be shortened and focused on the main results, that these patients delivered mostly by caesarean section, newborns are premature and small for gestational age, have lower Apgar score, and that dialysis is associated with a poor outcome.
Answer: Thank you for your recommendation; we shortened the manuscript and focus on the main results to be more precise; I rewrote the results and discussions section. (please see the attached manuscript). Lines 311-466.
Kindest regards